# The Role of Pharmacists in Minimizing the Risk Inherent in Unbundled Telehealth Services: A 12-Month Retrospective Case Study

**DOI:** 10.3390/pharmacy12060177

**Published:** 2024-11-25

**Authors:** Louis Talay, Matt Vickers, Tiffany Cheng

**Affiliations:** 1Faculty of Arts and Social Sciences, University of Sydney, Sydney, NSW 2050, Australia; 2Eucalyptus, Sydney, NSW 2000, Australia; matt@euclayptus.health (M.V.); tiffany.cheng@eucalyptus.health (T.C.)

**Keywords:** pharmacy, telehealth, error interception, prescriber error, sexual dysfunction

## Abstract

Pharmacists have often been viewed as the last line of defence against prescription errors in traditional care models. Although a large number of chronic care patients are using telehealth services to increase their access to continuous care, researchers have yet to investigate prescription safety in such settings in Australia. The absence of this literature is particularly concerning in the context of the Australian Government’s admission in a 2024 report that the national health system has not adequately addressed the World Health Organization’s ‘Medication without harm’ objective. One of the report’s key findings was that knowledge on digital direct-to-consumer services is insufficient. A defining feature of some of these services is their unbundling of the pharmacy component, which logically increases the risk for prescription errors. This study analyzed the frequency of which the Cloud pharmacy network intercepted prescription errors in an unbundled digital sexual dysfunction service for men. Investigators found that Cloud pharmacists were responsible for intercepting 22 (5.31%) the 414 prescribing errors observed in the Pilot Australia service in 2023, including 12 (8.05%) of the 149 prescription errors for premature ejaculation (PE) patients and 10 (3.77%) of the 265 errors for erectile dysfunction (ED) patients. Seven of the errors intercepted by Cloud pharmacists were of high or medium severity, including four drug contraindications, two cases of inadequate patient history reviews, and one case of inadequate counselling. This study also appears to be the first to provide digital prescribing error rate data in an Australian sexual healthcare setting, observing an error rate of 0.86% from 30,649 ED prescriptions, 1.13% from the 13,154 PE prescriptions, and a total prescription error rate of 0.95% (414 out of 43,792 prescriptions). These findings demonstrate the vital role of pharmacists in intercepting prescribing errors in unbundled telehealth services. Possible implications of these findings include the allocation of additional resources across the pharmacy sector and the establishment of regulatory safety standards for unbundled telehealth services.

## 1. Introduction

Technology is playing an increasingly important role in public healthcare systems throughout the world, including Australia. The most recent Australian Bureau of Statistics patient experience report revealed that 27.7% of the Australian population had at least one telehealth consultation in 2022–2023 and that people living with chronic health conditions and/or in areas of socioeconomic disadvantage were overrepresented in this statistic [1]. This trend observed in the latter two groups is arguably unsurprising given their greater access barriers to quality care and the potential of digital care modalities to overcome these barriers [2,3,4]. Whereas patients living in areas of socioeconomic disadvantage typically face the burden of having to undertake a significant geographical journey to access quality care, many patients with chronic conditions struggle to coordinate consultations across a team of specialists on an ongoing basis, which the nature of their illness necessitates [5]. Digital care models mitigate these access barriers by enabling patients to consult their clinicians at a time and place convenient to their circumstances. Other government publications in Australia have emphasized the important role telehealth can play in increasing access to care for stigmatized conditions, especially those affecting men and young adults [6,7]. Many patients with such conditions adopt passive coping strategies to conceal their perceived shame, which often exacerbate symptoms and give rise to comorbidities such as depression [8]. Although digital modalities cannot address the underlying social forces responsible for illness stigmatization, they allow the increasing large number of people who feel overwhelmed by face-to-face contact to access quality care [9,10].

To enhance care efficiency, many contemporary digital healthcare services unbundle their components [11]. Care models of this nature are particularly common for chronic, stigmatized conditions such as sexual dysfunction, for which specialist multidisciplinary care is required. Quality care for people living with sexual dysfunction (PWSD) necessitates, at the very least, a multidisciplinary team (MDT) consisting of a prescribing physician, a pharmacist, and a medical support officer to monitor and support the patient’s ongoing care journey. Although the continuous support function could theoretically be performed by the prescribing physician, this would likely exacerbate an already overburdened primary care system [12]. Medical support officers, therefore, play a crucial role in maintaining frequent and timely contact with patients whose conditions demand a high level of continuity, such as PWSD. Well-designed digital chronic care models automatically upload each patient–MDT communication to a central encrypted database to facilitate care coordination between each clinician.

Pilot is one Australia’s largest digital providers of sexual healthcare for men, offering treatment for erectile dysfunction (ED) and premature ejaculation (PE) [13]. Eligible patients are allocated an MDT that includes a prescribing doctor or nurse practitioner, a pharmacist, and a medical support officer. All MDT members are university-qualified clinicians and registered under the Australian Health Practitioner Regulation Agency [14]. Pilot medical support officers possess either a bachelor’s degree in pharmacy or nursing. Whereas prescribing physicians and medical support officers are employed directly by Eucalyptus (Pilot parent company) and have complete access to patient data through company’s central database on Metabase, pharmacists are sourced from an external third-party pharmacy network (Cloud Pharmacy) and can only view the medical information provided on patient scripts. The service’s dispensing model is detailed in the Section Section 2.

Although unbundling the pharmacy component of the Pilot ED and PE care model feasibly enhances its efficiency and specialization (rather than establishing a pharmacy network within the company), doing so may compromise its safety. In traditional care settings, pharmacists are often viewed as a last line of defence in preventing misprescription errors, and studies outside of Australia have demonstrated the significant rate at which they intercept such errors [15,16]. In emergency departments with handwritten prescriptions, for example, clinical pharmacists have been found to reduce prescribing errors by 76% (24.6% to 5.4%) [15]. While there is evidence that electronic prescribing reduces prescription error rates [17,18], pharmacists have still been found to intercept a significant number of errors in such prescribing models [16]. In Europe, this safety mechanism is often referred to as the integration of pharmaceutical validation within clinical decision support (CDS) systems [19,20]. A prospective multi-site analysis of pediatric patients in Spanish hospitals found that pharmacists intercepted 0.013 electronic prescribing errors per bed, per day [21]. A comparative study in a Belgian tertiary hospital setting revealed that pharmacists intervened in a significant number of electronic prescribing errors in both an offsite CDS and on-ward pharmacy models [19], Although the offsite CDS model (which mirrors the unbundled Pilot model) intercepted fewer errors (2.9%) than the on-ward model (13.3%), investigators explained that pharmacists in the latter group had more time and access to more information than those operating offsite. Earlier studies in Canadian, American, and British hospital settings have reported pharmacist intervention rates of 3.2, 7.8, and 8.5 percent, respectively [22,23,24]. Investigators across multiple studies have highlighted the important role pharmacists play in improving the quality and sophistication of electronic prescribing CDSs [25,26]. To the knowledge of the investigators, researchers are yet to explore error interception rates in unbundled digital prescribing services outside of hospital settings.

From a public health perspective, the importance of prescribing safety is arguably best evidenced by the World Health Organization’s (WHO) 2017 ‘Medication Without Harm’ patient safety challenge, which represented the third challenge of the organization’s history [27]. Medication errors can occur at one of two phases during the initiation of a medicated treatment plan: the prescribing phase or the dispensing phase [28]. A 2024 government report revealed that the WHO objective of “reducing patient harm generated by unsafe medication practices and medication errors” is yet to be adequately addressed in Australia. Among other things, the report underscored the failure of the current Australian health system to collect sufficient medication safety data, introduce industry-wide standards, and implement appropriate controls for direct-to-consumer communications [29]. Consistent with this report, peer-reviewed research on Australian digital prescribing safety appears to be confined to hospital settings [30,31]. Although a recent medication safety study was conducted on an unbundled digital weight-loss service in Australia, the study focused on dispensing errors and did not report prescribing error rates [28]. International data on prescribing safety also appear to be largely limited to hospital and community face-to-face settings [17]. The lack of medication safety research on unbundled digital chronic care models are a major concern for the increasingly large number of people who are subscribing to such services.

This study aims to analyze the frequency of PE and ED prescription errors in a cohort of Pilot Australia patients that were intercepted by the service’s external pharmacists. Although there is no available literature on prescriber error interception rates in traditional Australian pharmacy settings (let alone modern digital settings) for comparison, this study will generate valuable insights on the role of pharmacists in unbundled digital chronic care models.

## 2. Materials and Methods

### 2.1. Study Design

This study retrospectively analyzed all scripts for Pilot PE and ED patients that were sent to the Cloud pharmacy network for dispensing in 2023. The selection of this method was based on the UK National Health Service Health Research Authority’s “Defining Research table” by aligning with the following criteria [32]: “designed and conducted solely to define or judge current care or service”; “measures current service without reference to a standard”; “involves analysis of existing data”; and “patient/service users have chosen intervention independently of the service evaluation”. All patients consented to their de-identified data being used in this research. The Bellberry Human Ethics Committee approved this study on 22 November 2023 (No. 2023-05-563-A-1).

### 2.2. Programme Overview

The Pilot digital care service is accredited through the Australian Council on Healthcare Standards [33]. Pilot physicians assess responses to extensive pre-consultation questionnaires to determine patient eligibility for sexual health treatment. These questionnaires can contain over 100 questions and often include requests for clinical reports, tests results, and medical imaging. Upon determining patient eligibility, Pilot physicians forward PE and ED prescriptions to the Cloud Pharmacy network via email. Pharmacy staff then verify that the script adheres to Australian Regulatory Guidelines for Prescription Medicines before performing a review of the patient’s medical information. This clinical review includes an assessment of patient identity, allergies, potential contraindications and drug–drug interactions, dosage, treatment instructions, and any additional prescriber comments.

If pharmacists suspect an error has been made on a Pilot patient prescription, they raise their concern via a dedicated risk channel on Slack, San Francisco, Cal, USA—a cloud-based team communication platform with instant media sharing and voice and video call functionalities. Whenever pharmacists raise a concern, an alert is automatically sent to the Eucalyptus clinical auditing team (all qualified pharmacists or nurses), who then contact the patient’s prescriber. In cases they consider more serious, auditors will immediately set up a new consultation between the patient and their doctor. Cloud pharmacists cannot contact Pilot patients while a prescription is under review; however, they are free to provide counselling recommendations to Eucalyptus auditors, which they often include in their initial Slack message. If an identified error is resolved, the pharmacist will perform a final check of the medication, request patient payment, and dispense the order. Pharmacists are free to provide patients with their own relevant counselling via phone or SMS after orders are dispensed. Cloud pharmacy’s role in the Pilot sexual dysfunction service is displayed in Figure 1.

### 2.3. Sample

This study included all PE and ED prescriptions that were provided by Pilot Australia physicians between 1 January and 31 December 2023. These prescriptions included the following medications: Paroxetine, Sildenafil, Tadalafil, and Sertraline (all manufactured in Macquarie Park, NSW, Australia).

### 2.4. Procedures

To retrieve the errors identified by Cloud pharmacists, the investigators entered each of the four medications prescribed for Pilot PE and ED patients into the search feature of the dedicated risk channel on Slack (as individual terms). To determine the veracity of the identified errors, i.e., confirm whether errors identified by Cloud pharmacists were in fact errors, a qualified pharmacist from the Eucalyptus clinical auditing team compared Cloud pharmacist notes with corresponding patient data from the Pilot central data repository on Metabase (San Francisco, Cal, USA).

### 2.5. Endpoints

Coprimary endpoints were the ED and PE prescribing error rate of the Pilot service, and the proportion of these errors that were identified by Cloud pharmacists. Secondary endpoints included a percentage distribution of error type and severity and the actions taken by Pilot in response to medium and high severity errors identified by Cloud pharmacists.

### 2.6. Statistical Analysis

Descriptive data were reported in the total number of occurrences (percentages), mean scores, and standard deviation figures. A Braun and Clarke thematic analysis of pharmacist messages was conducted to determine the nature (type) of each error [34]. This method consists of a 6-step reflexive process involving iterations of codes and sub-themes to facilitate the development of themes or categories from large and complex qualitative data. Analysis of variance (ANOVA) tests were used to compare error rates across error types and severity levels. All quantitative analyses were conducted using RStudio, version 2023.06.1+524 (RStudio: Integrated Development Environment for R, Boston, MA, USA).

## 3. Results

A total of 43,792 ED and PE prescriptions were dispensed by the Cloud pharmacy network to Pilot Australia patients in 2023 (Table 1). Of these prescriptions, 414 (0.95%) contained an error, including 265 (0.86%) of the 30,649 ED prescriptions and 149 (1.13%) of the 13,154 PE prescriptions. Most errors (58%) were of low severity and pertained to inadequate medical history reviews of a non-critical nature (59.4%) and drug contraindications (20.77%).

Drug contraindication errors captured any case where a prescribing doctor had failed to detect a drug contraindication. Insufficient counselling errors were those in which scripts indicated a patient had not received their mandatory counselling for their treatment. Medical history check errors (critical) pertained to cases when prescribing doctors had not confirmed on a patient’s script that they had conducted a comprehensive review of an aspect of a patient’s medical history with potential for harm. Medical history check errors (non-critical) were the same as the former error but for aspects of a patient’s medical history with minimal risk potential. And finally, incorrect medication/dose errors referred to either a doctor prescribing the incorrect dose or medication or providing incorrect or unclear titration schedule instructions. Examples of the various error types and severities are presented in Table 2.

Cloud pharmacists were responsible for detecting 22 (5.31%) of these errors, including 12 for PE prescriptions and 10 for ED prescriptions (Table 3). Among these errors, 15 were deemed to have been of low severity, 5 of medium severity, and 2 of high severity. This study’s three investigators each identified the same five error types by phase 5 of the Braun and Clarke thematic analysis. During the final phase, the investigators settled on the wording of these five error types. The analysis found that, of the 22 confirmed errors, seven were related to drug contraindications, another seven to insufficient counselling, three to medical history checks (critical), three to medical history checks (non-critical), and two to dosing/medication errors. Multiple actions were taken in response to Cloud Pharmacy-detected errors but were aggregated as individual items to create a neater summary. The provision of extra patient counselling was the most common action, occurring in response to half (11) of all error cases, followed by a one-on-one performance review between a Pilot clinical auditor and the prescribing doctor (nine cases), cancelled orders (five cases), and review consultations between the MDT and patient (four cases).

Both high severity errors pertained to drug contraindications and resulted in cancelled orders and doctor performance reviews (Table 4). Among the five medium severity errors, two were drug contraindication errors that resulted in review consults, extra patient counselling, and a cancelled order; another two were medical history check errors that resulted in doctor performance reviews and a cancelled patient subscription; and one was an unattended counselling error resulting in extra patient counselling.

## 4. Discussion

To our knowledge, this was the first Australian study to investigate the frequency of prescription errors intercepted by an unbundled telehealth service’s external partner pharmacy. This study found that the Cloud pharmacy network reported 38 errors from the 43,792 ED and PE prescriptions issued by Pilot physicians in 2023, of which 22 were confirmed by a clinical auditor as legitimate errors. Although this finding represented a relatively low percentage (5.31%) of the total prescription errors detected in the Pilot sexual health service in 2023 (414), the fact that seven errors were considered to have been of high or medium severity and that drug contraindications were the most common error type highlights the added misprescription risk inherent in sexual dysfunction services that use unbundled digital care models. It is feasible that these medium–high severity errors could have resulted in serious patient harm had Cloud pharmacists failed to identify and report them prior to dispensing the relevant orders. No comparator is needed to assess the importance of intercepting these errors, given their potential for patient harm.

This study also appears to be the first to provide digital prescribing error rate data in an Australia sexual health service. A 2024 government assessment of the Australian health system’s implementation of WHO’s ‘Medication without harm’ strategy highlighted the dearth of medication safety data and its danger potential in direct-to-consumer services [27]. While the prescribing error rates of the Pilot ED (0.86%) and PE (1.13%) services appear reasonably low, they cannot be evaluated until data become available on comparable Australian telehealth programmes. The only other data of this description come from a recent study of an unbundled digital weight-loss service, which reported a prescribing error rate of 4.4% [35]. However, more data are needed to determine whether comparisons of different treatment models is appropriate. Nevertheless, findings from this study and that of the weight-loss service lay an important foundation for future comparisons and the development of more medication safety data across the Australian healthcare system. This study’s observation that most errors pertained to inadequate medical reviews of a non-critical nature (59.4%) and drug contraindications (20.77%) also suggest areas in which unbundled digital prescribing services and third-party pharmacists should dedicate additional safety resources.

Findings from this study could have several potential policy implications. Firstly, they will hopefully encourage stakeholders to conduct comparable analyses across the telehealth spectrum, which will enable a reassessment of Australia’s response to the WHO ‘medication without harm’ strategy. Secondly, they could prompt government consideration of regulatory standards for direct-to-consumer telehealth services—particularly in regard to clinical auditing processes. Thirdly, the study outcomes could possibly lead to further recognition of the vital role played by pharmacists in verifying prescriptions from any healthcare service. This recognition could result in the allocation of additional resources across the Australian pharmacy sector. And finally, on a global level, digital healthcare has been identified as a core component of each of the United Nations ‘Sustainable Development Goals’ to address inequalities. Findings from this study may engender discussion around the need for further support and investment for pharmacists working in digital services.

This study contained several limitations. Firstly, it collected pharmacy-reported errors from Slack messages, some of which contained grammatical errors and clunky phrasing, which could not be verified with the relevant pharmacist retrospectively. Secondly, this study was restricted to men and was thus unrepresentative of Australian society. Thirdly, this study only assessed male users of the Eucalyptus sexual health service and therefore cannot be generalized to other digital health services with unbundled care models. And finally, Eucalyptus clinical auditors (qualified pharmacists and nurses) were responsible for interpreting and coding all pharmacy-reported errors, increasing the likelihood of company performance bias.

Future research should seek to build upon this study’s foundational findings by assessing pharmacy-intercepted prescription errors in unbundled telehealth services for other complex conditions. Scholars should also consider analyzing prescription error rates in other telehealth services, qualitative studies that solicit pharmacist views on improving prescription safety in unbundled telehealth services, and comparative assessments of the role of pharmacists across a range of telehealth models.

## 5. Conclusions

This study generated vital foundational findings to the nascent field of unbundled telehealth service literature. It has been well established that telehealth models bring numerous access benefits, particularly for patients with stigmatized conditions. As a result of this knowledge, an increasing number of services are emerging for this kind of patient in the private sector, many of which unbundle their multidisciplinary components to enhance efficiency and specialization. Until this study, however, nothing was known about the degree to which unbundling pharmacies from medicated telehealth models affected prescription safety. This analysis revealed that a significant number of ED and PE prescriptions from the Pilot telehealth service contain errors and that the service’s third-party pharmacists play a vital role in detecting some of these errors.

## Figures and Tables

**Figure 1 pharmacy-12-00177-f001:**
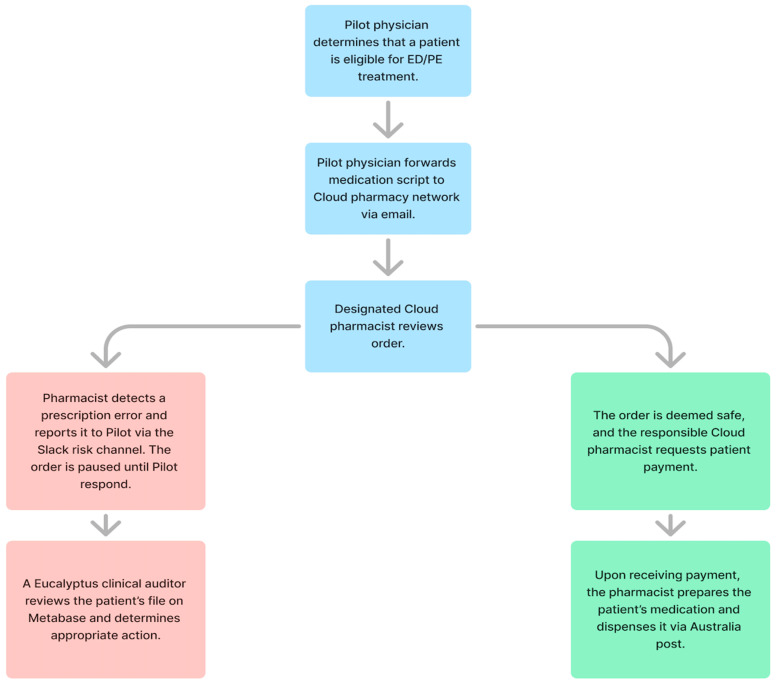
Cloud pharmacy role in Pilot PE and ED service.

**Table 1 pharmacy-12-00177-t001:** Total prescriptions and errors.

Prescriptions	No. (% of Total)
PE	13,154 (30)
ED	30,649 (70)
Total	43,792
**Total prescriptions with errors**	**No. (% of orders)**
PE	149 (1.13)
ED	265 (0.86)
Total	414 (0.95)
**Errors by severity**	**No. (% of total errors)**
High	62 (15)
Medium	112 (27)
Low	240 (58)
**Errors by type**	**No. (% of total errors)**
Inadequate medical history review (non-critical)	246 (59.4)
Drug contraindication	86 (20.77)
Inadequate medical history review (critical)	66 (15.94)
Insufficient counselling	11 (2.7)
Incorrect medication/dose	5 (1.21)

**Table 2 pharmacy-12-00177-t002:** Examples of Eucalyptus sexual health prescribing errors by error type and severity.

Condition	Error Type	Severity	Example
PE	Inadequate medical history review (non-critical)	Medium	Patient was recently diagnosed with cardiomyopathy. Practitioner acknowledged the diagnosis but solicited no further information before prescribing paroxetine.
		Low	Patient rated their mood as “not great” in the questionnaire, and, although the practitioner acknowledged this rating and provided mental health support resources, they did not reassess patient mood at 6-month consultation.
	Drug contraindication	High	Patient was prescribed paroxetine despite indicating that they were taking duromine, which is a moderate contraindication.
		Medium	Patient was taking rizatriptan for migraines (when required) and was prescribed paroxetine 10 mg, despite the contraindication risk of serotonin syndrome.
ED	Insufficient counselling	High	Patient revealed that they take recreational drugs including cocaine on a monthly basis. The practitioner did not inform the patient of the risks involved when combining such drugs with tadalafil.
		Medium	Patient was prescribed tadalafil 5 mg despite indicating that they were already taking a different brand of the medication. The practitioner provided no counselling on the risks of exceeding the recommended dose.
		Low	Patient was taking rosuvastatin but was prescribed sildenafil 100 mg, which has a moderate (although limited) severity interaction.
	Incorrect medication/dose	High	Patient was prescribed paroxetine instead of tadalafil, which was a high-risk prescribing event as the patient was already taking sertraline and therefore had the risk of serotonin toxicity syndrome.
		Medium	Patient was prescribed sertraline 100 mg daily instead of sildenafil 100 mg. Sertraline is a drug used to treat PE.
	Inadequate medical history review (non-critical)	Low	Patient indicated that they had a heart attack 3 years ago, but the practitioner did not address the stability of their condition during the consultation.

**Table 3 pharmacy-12-00177-t003:** Prescription errors intercepted by Cloud pharmacy.

Errors Reported by Cloud Pharmacy	No. (% of Total Prescriptions)
PE	22 (0.17)
ED	16 (0.05)
Total	38 (0.09)
**Errors confirmed by clinical auditor**	**No. (% of total errors)**
PE	12 (8.05)
ED	10 (3.77)
Total	22 (5.31)
**Errors by severity**	**No. (% of Cloud-detected errors)**
High	2 (9.1)
Medium	5 (22.7)
Low	15 (68.2)
**Errors by type**	**No. (% of Cloud-detected errors)**
Drug contraindication	7 (31.8)
Insufficient counselling	7 (31.8)
Inadequate medical history review (critical)	3 (13.6)
Inadequate medical history review (non-critical)	3 (13.6)
Incorrect medication/dose	2 (9.1)
**MDT/auditor response**	**No. (% of total responses)**
Extra patient counselling	11 (50)
Doctor performance review	9 (40.9)
Cancelled order	5 (22.7)
Review consult doctor–patient	4 (18.2)
Closed duplicate account	3 (13.6)
Cancelled patient subscription	1 (4.5)
Revise pharmacy instructions	1 (4.5)

**Table 4 pharmacy-12-00177-t004:** Pilot response to medium/high severity errors.

Condition	Error Type	Severity	Pilot MDT/Auditor Response
PE	Drug contraindication	Medium	- Review consult doctor–patient
ED	Medical history review (critical)	Medium	- Doctor performance review
ED	Medical history review (critical)	Medium	- Doctor performance review - Cancelled patient subscription
PE	Drug contraindication	Medium	- Review consult doctor–patient - Cancelled order
PE	Insufficient counselling	Medium	- Extra patient counselling
ED	Drug contraindication	High	- Doctor performance review - Cancelled order
ED	Drug contraindication	High	- Doctor performance review - Cancelled order

## Data Availability

The data presented in this study are available from the corresponding author upon reasonable request.

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
