# Peer review of "The Role of Pharmacists in Minimizing the Risk Inherent in Unbundled Telehealth Services: A 12-Month Retrospective Case Study"

_pharmacy, 2024, doi:10.3390/pharmacy12060177_

Round 1

Reviewer 1 Report

Comments and Suggestions for Authors

The study investigates pharmacist’s role in intercepting prescription errors within a telehealth service in Australia. The study specifically examines data from 2023, using records from third party partner that works with the telehealth service. The findings in the study demonstrate the vital role of pharmacists in intercepting prescribing errors in unbundled telehealth services.

Strengths:

The study supports the WHO’s ‘Medication without harm’ strategy, which emphasizes patient safety in prescription practices.

The methodology categorizes and analyze data effectively.

Weaknesses:

The data is not generalizable

There is lack in comparative findings

There is a potential bias

point-by-point list of major recommendations: 

Elaborate more on the prescription error interception with adding other studies to support the claim.

Elaborate more on how the findings could influence policies on pharmacist roles in digital health services and SDGs

Address the limitation of the study's focus on a specific demographic

point-by-point list of minor recommendations: 

In my opinion I recommend the following:
In the abstract include a statement on potential future implications?

In the introduction while I know that the study is focused on Australia but more information on a global level might enhance the paper.

I think that there may be more to the eye from this study interim of potential policy changes that could arise from this research

What is next please provide a point to future research needs such as exploring the role of pharmacists in different telehealth models

Author Response

point-by-point list of major recommendations: 

Comment 1:

Elaborate more on the prescription error interception with adding other studies to support the claim.

Response:

Thank you for this excellent suggestion. We have now added the following 13 lines (92-105) to the introduction, which reference 8 additional studies from other countries. The added text reads as follows:

“In Europe, this safety mechanism is often referred to as the integration of pharmaceutical validation within clinical decision support systems (CDS) [19,20]. A prospective multi-site analysis of pediatric patients in Spanish hospitals found that pharmacists intercepted 0.013 electronic prescribing errors per bed, per day [21]. A comparative study in a Belgian tertiary hospital setting revealed that pharmacists intervened a significant number of electronic prescribing errors in both an offsite CDS and on-ward pharmacy models [19], Although the offsite CDS model (which mirrors the unbundled Pilot model) intercepted fewer errors (2.9%) than the on-ward model (13.3%), investigators explained that pharmacists in the latter group had more time and access to more information than those operating offsite. Earlier studies in Canadian, American and British hospital settings have reported pharmacist intervention rates of 3.2, 7.8 and 8.5 percent, respectively [22-24]. Investigators across multiple studies have highlighted the important role pharmacists play in improving the quality and sophistication of electronic prescribing CDSs [25, 26].”

Here are the new citations:

  1. Lagreula, J., Maes, F., Wouters, D., et al. Optimizing pharmacists’ detection of prescribing errors: Comparison of on-ward and central pharmacy services. Journal of Clinical Pharmacy and Therapeutics, 2021, 46: 738-743
  2. Boussadi, A., Caruba, T., Karras, A., et al. Validity of a clinical decision rule-based alert system for drug dose adjustment in patients with renal failure intended to improve pharmacists’ analysis of medication orders in hospitals. International journal of Medical Informatics, 2013, 82:964-972
  3. Fernández-Llamazares, C., Pozas, M., Feal, B., et al. Profile of prescribing errors detected by clinical pharmacists in paediatric hospitals in Spain. International Journal of Clinical Pharmacy, 2013, 35:638-646
  4. Stasiak, P., Afilalo, M., Castelino, T., et al. Detection and correction of prescription errors by an emergency department pharmacy service. Journal of the Canadian Association of Emergency Physicians, 2014, 16: 193-206.
  5. Rothschild, J., Churchillm W., Erickson, A., et al. Medication errors recovered by emergency department pharmacists. Annals of Emergency Medicine, 2010, 55:513-521
  6. Abdel-Qader, D., Harper, L., Cantrill, J., et al. Pharmacists’ interventions in prescribing errors at hospital discharge. Drug Safety, 2010, 33:1027-1044.
  7. Beex-Osterhuis, M., de Vogelm E., van der Sijs, H., et al. Detection and correct handling of prescribing errors in Dutch hospital pharmacies using test patients. International Journal of Clinical Pharmacy, 2013, 35: 1188-1202.
  8. Estellat, C., Colombet, I., Vautier, S., et al. Impact of pharmacy validation in a computerized physician order entry context. International Journal for Quality in Health Care, 2007, 19: 317-325

Comment 2:

Elaborate more on how the findings could influence policies on pharmacist roles in digital health services and SDGs

Response:

Thank you for this important recommendation. We have now added an extra paragraph to the discussion section (lines 345-357) that summarizes the potential policy implications. This paragraph reads as follows:

Findings from the study could have several potential policy implications. Firstly, they will hopefully encourage stakeholders to conduct comparable analyses across the telehealth spectrum, which will enable a reassessment Australia’s response to the WHO ‘medication without harm’ strategy. Secondly, they could prompt government consideration of regulatory standards for direct-to-consumer telehealth services – particularly in regard to clinical auditing processes. Thirdly, the study outcomes could possibly lead to further recognition of the vital role played by pharmacists in verifying prescriptions from any healthcare service. This recognition could result in the allocation of additional resources across the Australian pharmacy sector. And finally, on a global level, digital healthcare has been identified as a core component of each of the United Nations ‘Sustainable Development Goals’ to address inequalities. Findings from this study may engender discussion around the need for further support and investment for pharmacists working in digital services.”       

Comment 3:

Address the limitation of the study's focus on a specific demographic

Response:

Thank you for this excellent suggestion. In addition to our second limitation, which highlights the study’s unrepresentativeness, we have now added the following limitation to the penultimate paragraph of the discussion section:

“Thirdly, the study only assessed male users of the Eucalyptus sexual health service and therefore cannot be generalized to other digital health services with unbundled care models.”

point-by-point list of minor recommendations: 

Comment 4:

In my opinion I recommend the following:
In the abstract include a statement on potential future implications?

Response:

Thank you for recognizing the importance of summarizing your previous recommendation in the abstract. We have now added the following sentence to the end of the abstract:

“Possible implications of these findings include the allocation of additional resources across the pharmacy sector and the establishment of regulatory safety standards for unbundled telehealth services.“

Comment 5:

In the introduction while I know that the study is focused on Australia but more information on a global level might enhance the paper.

Response:

Thank you for this insightful comment. We have now added the following 13 lines (92-105) to the introduction, which reference 8 additional studies from other countries. The added text reads as follows:

“In Europe, this safety mechanism is often referred to as the integration of pharmaceutical validation within clinical decision support systems (CDS) [19,20]. A prospective multi-site analysis of pediatric patients in Spanish hospitals found that pharmacists intercepted 0.013 electronic prescribing errors per bed, per day [21]. A comparative study in a Belgian tertiary hospital setting revealed that pharmacists intervened a significant number of electronic prescribing errors in both an offsite CDS and on-ward pharmacy models [19], Although the offsite CDS model (which mirrors the unbundled Pilot model) intercepted fewer errors (2.9%) than the on-ward model (13.3%), investigators explained that pharmacists in the latter group had more time and access to more information than those operating offsite. Earlier studies in Canadian, American and British hospital settings have reported pharmacist intervention rates of 3.2, 7.8 and 8.5 percent, respectively [22-24]. Investigators across multiple studies have highlighted the important role pharmacists play in improving the quality and sophistication of electronic prescribing CDSs [25, 26].”

Here are the new citations:

  1. Lagreula, J., Maes, F., Wouters, D., et al. Optimizing pharmacists’ detection of prescribing errors: Comparison of on-ward and central pharmacy services. Journal of Clinical Pharmacy and Therapeutics, 2021, 46: 738-743
  2. Boussadi, A., Caruba, T., Karras, A., et al. Validity of a clinical decision rule-based alert system for drug dose adjustment in patients with renal failure intended to improve pharmacists’ analysis of medication orders in hospitals. International journal of Medical Informatics, 2013, 82:964-972
  3. Fernández-Llamazares, C., Pozas, M., Feal, B., et al. Profile of prescribing errors detected by clinical pharmacists in paediatric hospitals in Spain. International Journal of Clinical Pharmacy, 2013, 35:638-646
  4. Stasiak, P., Afilalo, M., Castelino, T., et al. Detection and correction of prescription errors by an emergency department pharmacy service. Journal of the Canadian Association of Emergency Physicians, 2014, 16: 193-206.
  5. Rothschild, J., Churchillm W., Erickson, A., et al. Medication errors recovered by emergency department pharmacists. Annals of Emergency Medicine, 2010, 55:513-521
  6. Abdel-Qader, D., Harper, L., Cantrill, J., et al. Pharmacists’ interventions in prescribing errors at hospital discharge. Drug Safety, 2010, 33:1027-1044.
  7. Beex-Osterhuis, M., de Vogelm E., van der Sijs, H., et al. Detection and correct handling of prescribing errors in Dutch hospital pharmacies using test patients. International Journal of Clinical Pharmacy, 2013, 35: 1188-1202.
  8. Estellat, C., Colombet, I., Vautier, S., et al. Impact of pharmacy validation in a computerized physician order entry context. International Journal for Quality in Health Care, 2007, 19: 317-325

Comment 6:

I think that there may be more to the eye from this study interim of potential policy changes that could arise from this research

Response:

Thank you for this valuable recommendation. We have now added an extra paragraph to the discussion section (lines 345-357) that summarizes the potential policy implications. This paragraph reads as follows:

Findings from the study could have several potential policy implications. Firstly, they will hopefully encourage stakeholders to conduct comparable analyses across the telehealth spectrum, which will enable a reassessment Australia’s response to the WHO ‘medication without harm’ strategy. Secondly, they could prompt government consideration of regulatory standards for direct-to-consumer telehealth services – particularly in regard to clinical auditing processes. Thirdly, the study outcomes could possibly lead to further recognition of the vital role played by pharmacists in verifying prescriptions from any healthcare service. This recognition could result in the allocation of additional resources across the Australian pharmacy sector. And finally, on a global level, digital healthcare has been identified as a core component of each of the United Nations ‘Sustainable Development Goals’ to address inequalities. Findings from this study may engender discussion around the need for further support and investment for pharmacists working in digital services.”       

Comment 7:

What is next please provide a point to future research needs such as exploring the role of pharmacists in different telehealth models

Response:

Thank for this important suggestion. We have now added the suggestion to the final paragraph of the discussion section. This paragraph reads as follows:

Future research should seek to build upon this study’s foundational findings by assessing pharmacy-intercepted prescription errors in unbundled telehealth services for other complex conditions. Scholars should also consider analyzing prescription error rates in other telehealth services, qualitative studies that solicit pharmacist views on improving prescription safety in unbundled telehealth services, and comparative assessments of the role of pharmacists across a range of telehealth models.”

Reviewer 2 Report

Comments and Suggestions for Authors

Thank you very much for giving an opportunity to review the present manuscript. The authors examined the frequency in which the Cloud pharmacy network intercepted prescription errors in an unbundled digital sexual dysfunction service for men. This study is the first to provide data on digital prescribing error rates in a non-hospital setting in Australia and demonstrates the important role pharmacists play in detecting prescribing errors in unbundled telehealth services. The manuscript is well-written and translated. I did not find any major flaws.

However, there are a few minor issues listed below:

I recommend illustrating the unbundled telehealth service model in this study with a more comprehensible schematic.

In Figure 1, the text in the red background section is cut off halfway.

In Table 1, under "errors by severity," the total adds up to 416. Shouldn't the total be 414?

L209-218: How about creating a separate table with specific examples?

Author Response

Thank you very much for giving an opportunity to review the present manuscript. The authors examined the frequency in which the Cloud pharmacy network intercepted prescription errors in an unbundled digital sexual dysfunction service for men. This study is the first to provide data on digital prescribing error rates in a non-hospital setting in Australia and demonstrates the important role pharmacists play in detecting prescribing errors in unbundled telehealth services. The manuscript is well-written and translated. I did not find any major flaws.

However, there are a few minor issues listed below:

Comment 1:

・I recommend illustrating the unbundled telehealth service model in this study with a more comprehensible schematic.

Response:

Thank you very much for noticing the vagueness of Figure 1. We have now replaced this Figure with one that contains an extra text box on the right to clarify a pharmacist’s actions after they deem an order to be safe. The new Figure also completes the sentence that was cut out of the higher red box in the previous figure and improves the clarity of the text in the first blue box.

Comment 2:

・In Figure 1, the text in the red background section is cut off halfway.

Response:

Thank you once again for detecting this oversight. As explained in the response to comment 1, we have now addressed this issue.

Comment 3:

・In Table 1, under "errors by severity," the total adds up to 416. Shouldn't the total be 414?

Response:

Thank you very much for noticing this. We miscounted the number of low errors and have now inserted the correct figure (240 instead of 242). The total error count is now consistent across the table.

Comment 4:

・L209-218: How about creating a separate table with specific examples?

Response:

Thank you for this excellent suggestion. We have now added a new table after the second paragraph of the results section (line 243) that provides examples of prescribing errors across conditions, error types and severity levels. Here is a copy of the table:

Table 2: Examples of Eucalyptus sexual health prescribing errors by error type and severity

Condition

Error type

Severity

Example

PE

Inadequate medical history review (non-critical)

Medium

Patient was recently diagnosed with cardiomyopathy. Practitioner acknowledged the diagnosis but solicited no further information before prescribing paroxetine.

Low

Patient rated their mood as "not great" in the questionnaire and although the practitioner acknowledged this and provided mental health support resources, they did not reassess patient mood at 6-month consultation.

Drug contraindication

High

Patient was prescribed paroxetine despite indicating that they were taking duromine, which is a moderate contraindication.

Medium

Patient was taking rizatriptan for migraines (when required) and was prescribed paroxetine 10mg, despite the contraindication risk of serotonin syndrome.

ED

Insufficient counselling

High

Patient revealed they take recreational drugs including cocaine on a monthly basis. The practitioner did not inform the patient of the risks involved when combining such drugs with Tadalafil.

Medium

Patient was prescribed tadalafil 5mg despite indicating that they were already taking a different brand of the medication. The practitioner provided no counselling on the risks of exceeding the recommended dose.

Low

Patient was taking rosuvastatin, but was prescribed sildenafil 100mg, which has a moderate (although limited) severity interaction.

Incorrect medication/dose

High

Patient was prescribed paroxetine instead of tadalafil, which was a high risk prescribing event as the patient was already taking sertraline and therefore had risk of serotonin toxicity syndrome.

Medium

Patient was prescribed sertraline 100mg daily instead of sildenafil 100mg. Sertraline is a drug used to treat PE.

Inadequate medical history review (non-critical)

Low

Patient indicated they had a heart attack 3 years ago, but the practitioner did not address the stability of their condition during the consultation.

Round 2

Reviewer 1 Report

Comments and Suggestions for Authors

Dear Author

Thank you so much for reflecting the comments